# Searching for Stage-wise Neural Graphs In the Limit

## Abstract

Search space is a key consideration for neural architecture search. Recently, [32] found that randomly generated networks from the same distribution perform similarly, which suggests we should search for random graph distributions instead of graphs. We propose graphon as a new search space. A graphon is the limit of Cauchy sequence of graphs and a scale-free probabilistic distribution, from which graphs of different number of nodes can be drawn. By utilizing properties of the graphon space and the associated cut-distance metric, we develop theoretically motivated techniques that search for and scale up small-capacity stage-wise graphs found on small datasets to large-capacity graphs that can handle ImageNet. The scaled stage-wise graphs outperform DenseNet and randomly wired Watts-Strogatz networks, indicating the benefits of graphon theory in NAS applications.

## 1 Introduction

Neural architecture search (NAS) aims to automate the discovery of neural architectures with high performance and low cost. Of primary concern to NAS is the design of the search space [23], which needs to balance multiple considerations. For instance, too small a space would exclude many good solutions, whereas a space that is too large would be prohibitively expensive to search through. An ideal space should have a one-to-one mapping to solutions and sufficiently smooth in order to accelerate the search.

A common technique [37, 17, 35, 19, 24, 34] to keep the search space manageable is to search for a small cell structure, typically containing about 10 operations with 1-2 input sources each. When needed, identical cells are stacked to form a large network. This technique allows cells found on, for instance, CIFAR-10 to work on ImageNet. Though this practice is effective, it cannot be used to optimize the overall network structure.

In both manual and automatic network design, the overall network structure is commonly divided into several stages, where one stage operates on one spatial resolution and contains several near-identical layers or multi-layer structures (i.e., cells). For example, ResNet-34 [11] contains 4 stages with 6, 8, 12, and 6 convolutional layers, respectively. DenseNet-121 [12] contains 4 stages with 6, 12, 24, and 16 two-layer cells. AmoebaNet-A [24] has 3 stages, within each 6 cells are arranged sequentially. Among cells in the same stage, most connections are sequential with skip connections occasionally used. As an exception, DenseNet introduces connections between every pairs of cells within the same stage.

Here we emphasize the difference between a stage and a cell. A cell typically contains about 10 operations, each taking input from 1-2 other operations. In comparison, a stage can contain 60 or more operations organized in repeated patterns and the connections can be arbitrary. A network usually contains only 3-4 stages but many more cells. In this paper, we focus on the network organization at the level of stage rather than cell.

[32] recently showed that the stage structure can be sampled from probabilistic distributions of graphs, including Erdős-Rényi (ER) (1960), Watts-Strogatz (WS) (1998), and Barabási-Albert (BA) (1999), yielding high-performing networks with low in-group variance. This finding suggests the random graph distribution, rather than the exact graph, is the main causal factor behind network performance. Thus, searching for the graph is likely not as efficient as searching for the random

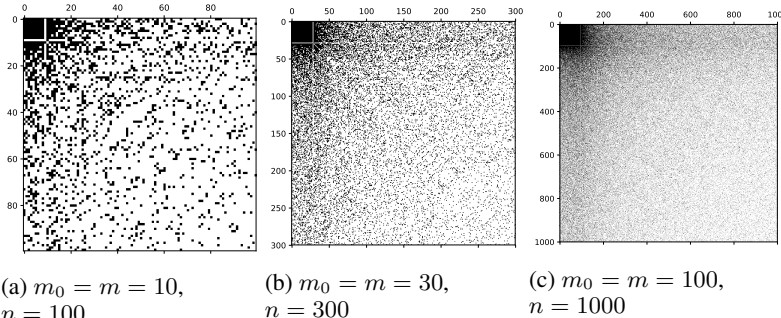

(a) $m_0 = m = 10$, $n = 100$

(b) $m_0 = m = 30$, $n = 300$

(c) $m_0 = m = 100$, $n = 1000$

Figure 1: Three adjacency matrices of graphs generated by the Barabási-Albert model with $m = m_0 = 0.1n$. A black dot at location $(i, j)$ denotes an edge from node $i$ to node $j$. The sequence of matrices converges to its limit, the graphon, as $n \to \infty$.

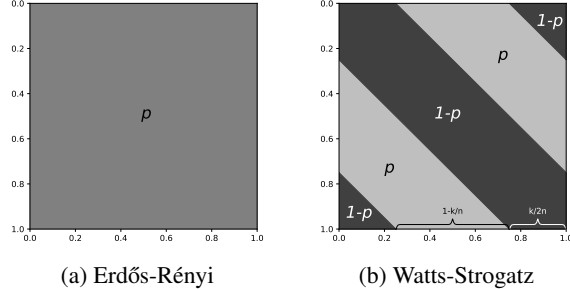

(a) Erdős-Rényi

(b) Watts-Strogatz

Figure 2: Graphons for common random graph models. Different shades denote different probabilities (e.g., $p$ and $1 - p$). The Erdős-Rényi model has two parameters: number of nodes $n$ and probability $p$. The Watts-Strogatz (WS) model has three parameters: number of nodes $n$, replacement probability $p$, and initial neighborhood width $k$. Technically, the WS model has a constant number of edges, violating exchangeability for random graphs; graphs sampled from (b) converges in probability to the same number of edges as $n$ increases.

graph distribution. The parameter space of random graph distributions may appear to be a good search space.

We propose a different search space, the space of graphons [20], and argue for its superiority as an NAS search space. Formally introduced in Section 3, a graphon is a measurable function defined on $[0, 1]^2 \to [0, 1]$ and a probabilistic distribution from which graphs can be drawn. Graphons are limit objects of Cauchy sequences of finite graphs under the cut distance metric.

Figure 1 visualizes three adjacency matrices randomly generated by the Barabási-Albert (BA) model with increasing numbers of nodes. It is easy to see that, as the number of nodes increases, the sequence of random graphs converges to its limit, a graphon. The BA model starts with an initial seed graph with $m_0$ nodes and arbitrary interconnections. Here we choose a complete graph as the seed. It sequentially adds new nodes until there are $n$ nodes in the graph. For every new node $v_{new}$, $m$ edges are added, with the probability of adding an edge between $v_{new}$ and the node $v_i$ being proportional to the degree of $v_i$. In Figure 1, we let $m = m_0 = 0.1n$. The fact that different parameterization results in the same adjacency matrix suggests that directly searching in the parameter space will revisit the same configuration and is less efficient than searching in the graphon space.

Additionally, graphon provides a unified and more expressive space than common random graph models. Figure 2 illustrates the graphons for the WS and the ER models. We can observe that these random models only capture a small proportion of all possible graphons. The graphon space allows new possibilities such as interpolation or striped combination of different random graph models.

Finally, graphon is *scale-free*, so we should be able to sample an arbitrary-sized stage-wise architecture with identical layers (or cells) from a graphon. This allows us to perform expensive NAS

on small datasets (e.g., CIFAR-10) using low-capacity models and obtain large stage-wise graphs to build large models. By relating graphon theory to NAS, we provide theoretically motivated techniques that scale up stage-wise graphs, which are shown to be effective in practice.

Our experiments aim to fairly compare the stage-wise graphs found by our method against DenseNet and the WS random graph model by keeping other network structures and other hyperparameters constant. The results indicate that the graphs found outperform the baselines consistently across a range of model capacities.

The contribution of this paper revolves around building a solid connection between theory and practice. More specifically,

- We propose graphon, a generalization of random graphs, as a search space for stage-wise neural architecture that consists of connections among mostly identical units.

- We develop an operationalization of the theory on graphon in the representation, scaling and search of neural stage-wise graphs that perform well in fair comparisons.

## 2    RELATED WORK

We review the NAS literature with a focus on the distinction between cell and stage structures. In the pioneering work of [36], a recurrent neural network served as the controller that outputs all network parameters for all layers without such distinction. Later works attempted to reduce the cost of search by constraining the search to cell structures only. [35] searched for a single cell structure and perform downsampling using pooling operations. [37], [33], and [24] searched for two types of cells: a reduction cell that includes downsampling, and a normal cell that does not. [17] grew the cell structure from the simplest 1-operation cell to the maximum of 5 operations. DARTS [19] relaxed discrete choices and enabled gradient-based optimization. [5] imposed a probabilistic formulation on DARTS.

While various manual designs of stage structures have been proposed (e.g., Huang et al. 12, Larsson et al. 16), NAS for stage-wise graph has received relatively little attention. [32] found that random graphs generated by properly parameterized Watts-Strogatz models outperform manual designs for stage-wise structures. [30] redefined the stage-wise graphs so that a node can take multiple inputs but output only a single channel, resulting in large stage-wise graphs with up to 2560 nodes. [27] evolved the connections between multiple residual blocks for applications in video processing. As a differennt type of global structure, [18] optimized organization of downsampling and upsampling. In summary, we believe that NAS for stage structures is an emerging research direction with many unexplored opportunities.

Another approach for accelerate search is to share weights among different architectures. [28] built a lattice where a chain-structured network is a path from the beginning to the end. In ENAS [22], a controller is trained using gradient policy to select a subgraph, which is subsequently trained using cross-entropy. The process repeats, sharing network weights across training sessions and subgraphs. In the one-shot approach [4, 2, 6], the hypergraph is only trained once. After that, subgraphs are sampled from the hypergraph and evaluated. Finally, the best performing subgraph is retrained. Our focus in this paper is to validate that the mathematical theory of graphon can be effectively operationalized and leave weight sharing as future work.

## 3    BACKGROUND ON GRAPHON

**Definition 1.** *A graphon is a symmetric, measurable function* $W : [0, 1]^2 \to [0, 1]$.

The definition of graphon is straightforward, but its relation with graphs requires some standard concepts from real analysis, which we introduce below. In machine learning, graphon has found applications in hierarchical clustering [8] and graph classification [10].

### 3.1 PRELIMINARIES

**A metric space** is a space with a distance function. More specifically, a metric space is a set $\mathbb{M}$ with a metric function $d : \mathbb{M} \times \mathbb{M} \to \mathbb{R}$ that defines the distance between any two points in $\mathbb{M}$ and satisfies the following properties: (1) Identity: $d(x, x) = 0$. (2) Symmetry: $d(x, y) = d(x, y)$. (3) Triangular inequality: $d(x, y) \leq d(x, z) + d(z, y)$. As an example, the set of real numbers $\mathbb{R}$ with the absolute difference metric $d_{abs}(x, y) = |x - y|$ is a metric space.

**A Cauchy sequence** is defined as a sequence $(x_1, x_2, \ldots)$ whose elements become infinitely close to each other as we move along the sequence. Formally, for any positive $\epsilon \in \mathbb{R}_+$, there exists a positive integer $N \in \mathbb{Z}_+$ such that $d(x_j, x_i) < \epsilon, \forall i, j > N, i, j \in \mathbb{Z}_+$.

**A complete metric space** is a metric space $(\mathbb{M}, d)$ in which every Cauchy sequence converges to a point in $\mathbb{M}$. It turns out that some familiar spaces, such as rational numbers $\mathbb{Q}$ under the metric $d_{abs}$, are not complete. As an example, the sequence defined by $x_0 = 1, x_{n+1} = x_n/2 + 1/x_n$ converges to an irrational number $\sqrt{2}$. The space of real numbers $\mathbb{R}$ with $d_{abs}$, however, is a complete metric space. Given any metric space, we can form its completion, in general, by adding limit points. In the case of graphs, the limits allow explicit descriptions.

### 3.2 GRAPHON, THE CUT DISTANCE, AND DIGRAPHON

We consider a weighted undirected graph $\mathcal{G} = (\mathbb{V}, \mathbb{E})$ where every $v_i \in \mathbb{V}$ is associated with a node weight $\alpha_i$ and every edge $e_{ij} \in \mathbb{E}$ is associated with edge weight $\beta_{ij}$. When necessary, we also write $\alpha_i = \alpha_i(\mathcal{G})$ to highlight the graph the node belongs to. The weighted graph is a generalization of the simple graph, where all nodes have weight 1 and all edge have weight 1. Edges that do not exist are considered to have weight 0. [3] show the following.

**Theorem 1.** *Every Cauchy sequence of weighted graphs in the metric $\delta_\square$ converges to a graphon.*

**Theorem 2.** *An weighted graphon is the limit of some Cauchy sequence in the metric $\delta_\square$.*

The definition of the cut distance $\delta_\square$ relies on its discrete versions $d_\square$ and $\hat{\delta}_\square$. To avoid unnecessary technical details, here we introduce the intuition of $d_\square$ and leave $\hat{\delta}_\square$ and $\delta_\square$ to Appendix C.

For a partition $\mathbb{S}, \mathbb{T}$ of $\mathbb{V}$, the cut size is defined as $\text{cut}(\mathbb{S}, \mathbb{T}, \mathcal{G}) = \sum_{v_i \in \mathbb{S}, v_j \in \mathbb{T}} \alpha_i \alpha_j \beta_{ij}$. When two graphs $\mathcal{G}$ and $\mathcal{G}' = (\mathbb{V}', \mathbb{E}')$ have the same set of nodes (i.e. $\mathbb{V} = \mathbb{V}'$), the partition $\mathbb{S}, \mathbb{T}$ applies to both graphs. We can then define

$$d_\square(\mathcal{G}, \mathcal{G}') = \max_{\mathbb{S}, \mathbb{T} \subset \mathbb{V}} \frac{1}{|\mathbb{V}|^2} |\text{cut}(\mathbb{S}, \mathbb{T}, \mathcal{G}) - \text{cut}(\mathbb{S}, \mathbb{T}, \mathcal{G}')| \tag{1}$$

The seemingly counter-intuitive $d_\square$ is a sensible metric for random graphs. Consider two random graphs with $n$ nodes from the same Erdős-Rényi model with edge density $1/2$. As they are identically distributed, their distance should be small. However, their edit distance, or the number of edges where the two graphs differ, is likely quite large. In contrast, $d_\square$ is only $O(1/n)$ with high probability, which is consistent with our intuition.

For directed graphs with the potential of self-loops, [7] define **a digraphon** as a 5-tuple $(W_{00}, W_{01}, W_{10}, W_{11}, w)$. For nodes $u_i$ and $u_j$, $W_{00}(i, j)$ describes the probability that no edge exists between them; $W_{01}(i, j)$ the probability that an edge goes from $u_i$ to $u_j$; $W_{10}(i, j)$ the probability that an edge goes from $u_j$ to $u_i$, and $W_{11}(i, j)$ the probability that two edges go both ways. $w_i$ is the probability for a self-loop at $u_i$. For all $i, j$, $W_{00}(i, j) + W_{01}(i, j) + W_{10}(i, j) + W_{11}(i, j) = 1$.

## 4 APPROACH

Since the graphon is a limit object, we must approaximate it with finite means. Here we use a step function approximation. We utilize a matrix $B \in [0, 1]^{n \times n}$ as the adjacency matrix of a weighted graph whose edgeweights $\beta_{ij}$ represent the mean of a $\frac{1}{n} \times \frac{1}{n}$ square from the graphon. Its nodeweights are uniformly $\frac{1}{n}$. This approximation converges to the graphon when $n$ tends to infinity (Lemma 3.2, Borgs et al. 3). In order to represent directed acyclic graphs in neural networks, we require the adjacency matrices to be upper-triangular and has a zero vector on the diagonal. This

is equivalent to imposing a total ordering $\prec$ on the nodes and requiring $i \prec j$ for a directed edge to go from $v_i$ to $v_j$.

In the following, we propose theoretically motivated techniques for sampling stage-wise graphs of different sizes from the graphon representation and NAS for graphon.

## 4.1 SAMPLING FROM AND SCALING A GRAPHON

Given a finite graph $\mathcal{G}_{est}$ with edge weights $\beta_{ij} \in [0, 1]$ and uniform nodeweights, we can sample a simple graph $\mathcal{G}_{sample}$ (i.e. with all edgeweights equal to 0 or 1) with $n$ nodes by drawing every edge independently as a 0-1 variable from the Bernoulli distribution parameterized by $\beta_{ij}$. [3] show $\mathcal{G}_{sample}$ converges to $\mathcal{G}_{est}$ in probability as the number of nodes $n$ increases.

$$\Pr\left(d_\Box(\mathcal{G}_{est}, \mathcal{G}_{sample}) < \frac{4}{\sqrt{n}}\right) > 1 - 2^{-n} \tag{2}$$

This procedure requires $\mathcal{G}_{est}$ and $\mathcal{G}_{sample}$ to have the same number of nodes.

By utilizing properties of the graphon metric space, we can sample a graph with more than $n$ nodes from the graph $\mathcal{G}_{est}$ with $n$ nodes. When the upsampling factor is an integer $k$, we first create a new weighted graph $\mathcal{G}_{est}[k]$ with $kn$ nodes using the so-called k-fold blow-up procedure, and then use the above procedure to sample $\mathcal{G}_{sample}$ with $kn$ nodes. The *k-fold blow-up* of a graph $\mathcal{G}_{est}$ is created by splitting every node in $\mathcal{G}_{est}$ into $k > 1$ nodes and connect the new nodes if and only if the original nodes are connected. More formally, for any two nodes $v_i$ and $v_j$, we create $k$ copies $v_{i1}, \ldots, v_{ik}$ and $v_{j1}, \ldots, v_{jk}$. The new graph $\mathcal{G}_{est}[k]$ contains the edge from $v_{ip}$ to $v_{jq}$ if and only if there is an edge from $v_i$ to $v_j$ in the original graph. This scheme is justified as the cut distance between a graph $\mathcal{G}$ and its k-fold blow-up $\mathcal{G}[k]$ is zero (See Corollary 2 in Appendix D).

$$\delta_\Box(\mathcal{G}, \mathcal{G}[k]) = 0 \tag{3}$$

To handle the case when the upsampling factor is not an integer, we propose a method called *fractional blow-up*. Suppose we want to create a stage-wise graph with $kn + m$ nodes. We first perform k-fold blow-up to create a new graph with $kn$ nodes and equal nodeweight $1/kn$. After that, we shift the nodeweights such that the first $m$ nodes have nodeweights $2/(kn+m)$ and the rest $kn - m$ nodes have nodeweights $1/(kn+m)$. We subsequently split the first $m$ node into 2 nodes, yielding a graph of equal nodeweights. As detailed in Appendix D, the cut distance between $\mathcal{G}_{est}$ and $\mathcal{G}'_{est}$ can be bounded by

$$\delta_\Box(\mathcal{G}_{est}, \mathcal{G}'_{est}) \leq \beta_\Delta \frac{(kn - m)m}{kn(kn + m)}. \tag{4}$$

where $\beta_\Delta$ denotes the maximum difference between any two edge weights. Since $\beta_{ij} \in [0, 1]$, $\beta_\Delta \leq 1$. From $\mathcal{G}'_{est}$, we can then sample a simple graph with $kn + m$ nodes.

It is worth noting that the proposed upscaling methods differ from conventional upsampling techniques like linear or bilinear interpolation. In Appendix D.3, we show that, under moderate conditions, the k-fold blow-up graph is strictly closer to the original graph than a graph created by interpolation.

## 4.2 SEARCHING FOR AN ADJACENCY MATRIX

We now introduce the search algorithm. When optimizing discrete objects with gradient descent, the Gumbel softmax [14, 31] has been widely used. Given a multinomial distribution with probabilities $\pi_0, \ldots, \pi_K$, we independently draw a $K$-dimensional vector $\boldsymbol{\beta}$ from a Gumbel distribution: $\forall k, \gamma_k \sim \text{Gumbel}(0, 1)$. The Gumbel softmax is defined as

$$a_k = \frac{\exp\left((\log \pi_k + \gamma_k)\tau^{-1}\right)}{\sum_{i=1}^{K} \exp\left((\log \pi_i + \gamma_i)\tau^{-1}\right)} \tag{5}$$

The Gumbel softmax can be understood as pitting the choices from 1 to $K$ against each other, while the random perturbation from $\boldsymbol{\beta}$ enables exploration.

Our search algorithm optimizes the input connections to each node in the stage-wise graph. Empirically, we find it important to let different cells in the same stage learn to collaborate during the

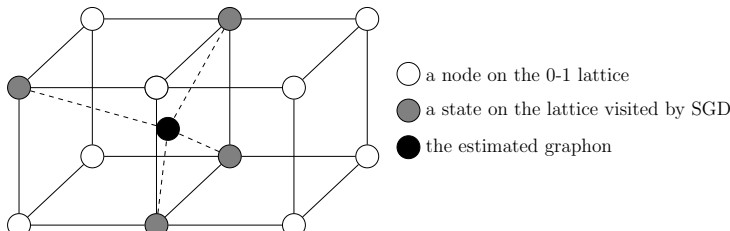

Figure 3: The intuition on the search for the optimal graphon. When searching in the space of adjacency matrices, we only can only move on the 0-1 lattice. The optimal graphon, denoted by the filled black dot, is estimated by taking the average of the states visited by SGD.

search. Therefore, we devise an algorithm that pits different input *combinations* against each other. Specifically, the node $v_i$ may take input from nodes $v_1, \ldots, v_{i-1}$, from which we sample $K$ subsets. For example, for the seventh node $v_7$, we could sample $\{v_1, v_3, v_5\}$ or $\{v_5, v_6\}$ and so on. We assign a structural parameter $\pi_k$ to every input subset. Finding the adjacency matrix amounts to finding the best input subset for every node. We also assign separate model parameters to the same node in different input subsets. This allows all nodes in the same subset to learn to collaborate, which would be difficult if the parameters were shared across input subsets. The outputs of nodes in the same subset are aggregated using concatenation with zero padding for aligning the dimensions. During the search, the input to node $v_i$ is computed as the convex combination of the outputs from all $K$ subsets, $\sum_{k=1}^{K} a_k \, out(k)$. For the pseudo-code of the search algorithm, the reader is referred to Appendix B.

We use cross-entropy loss and standard stochastic gradient descent (SGD) with momentum to optimize the model parameters together with subset weights $\boldsymbol{\pi}$. After the search completes, We pick the input subset with the highest $\pi_k$ in the final adjacency matrix.

However, the goal of the search is to find a probabilistic distribution of random graphs, not a single graph. As indicated by [21] and [13], the last phase of SGD may be considered as a Markov chain that draws samples from a stationary distribution. Figure 3 illustrates this intuition. Thus, we compute the graphon as the average of adjacency matrix found by the last phase of the search.

## 5 EXPERIMENTS

### 5.1 SETUP

DenseNet [12] is a classical example where the stage-wise structure plays a major role in improving the network performance. Therefore, in the experiment, we focus on improving and comparing with DenseNet. In order to create fair comparisons, we focus the search on the stage-wise structure and strictly follow DenseNet for the global network layout and the cell structure (See details in Appendix A.1).

Following [32], we start the search from the graphon that corresponds to the WS distribution with $k/n = 0.4$ and $p = 0.75$, which we denote as WS-G(0.4, 0.75). We limit the search to input subsets that differ by 1 edge from the starting point. We perform the search on CIFAR-10 for 300 epochs with 8 cells in every stage. The search completes within 24 hours on 1 GPU.

We create four groups of comparison for the four DenseNet variants: DenseNet-121, DenseNet-169, DenseNet-201, and DenseNet-264 in an increasing order of model capacity. We use the proposed scaling method (Section 4.1) to scale up the graphs in the four stages to roughly match the corresponding DenseNet variant. The largest stage-wise graph, containing 64 nodes, is used in the DenseNet-264 group. We also adjust the growth rate $c$ so that the numbers of parameters of all models are as close to DenseNet as possible to enable fair comparisons. However, as the number of parameters depends on the stage-wise connections, which are randomly drawn from the same distribution, we do not have precise control over the number of parameters. We strictly follow the standard hyperparameters and data augmentation techniques used by DenseNet. The networks are

trained for 90 epoch on ImageNet; results from 6 independent training sessions are averaged. For a detailed list of hyperparameters and data augmentation, see Appendix A.2.

We introduce two other baseline models besides DenseNet. In the first model, the stage-wise graphs are generated by randomly deleting edges from the fully connected graph of DenseNet. In the second model, we use random graphs generated from WS-G(0.4, 0.75), which are similar to the best random architecture in [32] and shown to be competitive with Amoeba [24], PNAS [17] and DARTS [19] on an equal-parameter basis.

## 5.2 RESULTS

For evaluation, we report the average performance on the ILSVRC-2012 validation set [26] and the new ImageNet V2 test set [25]. For ILSVRC-2012 validation, we report the performance directly after epoch 90. For ImageNet V2, we select the best performing model on the ILSVRC-2012 validation set from the 90 epochs and test it on ImageNet V2. In order to mitigate the effects of random variance, we sample 6 graphs from every graphon and report the average accuracy and standard deviation, as well as the number of parameters in Table 1.

Across all groups of comparison and model parameter setting, the graphon found by our algorithm consistently achieves the highest accuracy despite having the least parameters. On ImageNet validation, the top-1 performance gap between our method and DensetNet is 0.4% except for 0.1% for DenseNet-121. On ImageNet V2, the performance gap is up to 0.8%. The WS-G baseline in most comparisons is stronger than DenseNet but weaker than the propose technique.

## 5.3 DISCUSSION

We attribute the performance differences to the stage-wise graphs, since we have strictly applied the same setting, including the global network structure, the cell structure, and hyperparameter settings.

The first conclusion we draw is the effectiveness of the theoretically motivated scaling technique for graphon. We scaled up the 11-node graph found by the search to graphs with up to 64 nodes in the experiments. We also scaled the WS(4, 0.25) network, initially defined for 32 nodes in [32], to 64 nodes in the DenseNet-264 group. The experiments show that after scaling, the relative rankings of these methods are maintained, suggesting that the proposed scaling technique incurs no performance loss.

Second, we observe the standard deviations for most methods are low, even though they edge a bit higher for ImageNet V2 where model selection has been carried out. This is consistent with the findings of [32] and reaffirms that searching for random graphs is a valid approach for NAS.

Finally, we emphasize that these results are created for the purpose of fair comparisons and not for showcasing the best possible performance. Our goal is to show that the graphon space and the associated cut distance metric provide a feasible approach for NAS and the empirical evidences support our argument.

## 6 CONCLUSIONS

The design of search space is of paramount importance for neural architecture search. Recent work [32] suggests that searching for random graph distributions is an effective strategy for the organization of layers within one stage. Inspired by mathematical theories on graph limits, we propose a new search space based on graphons, which are the limits of Cauchy sequences of graphs based on the cut distance metric.

The contribution of this paper is the operationalization of the graphon theory as practical NAS solutions. First, we intuitively explain why graphon is a superior search space than the parameter space of random graph models such as the Erdős-Rényi model. Furthermore, we propose a technique for scaling up random graphs found by NAS to arbitrary size and present a theoretical analysis under the cut distance metric associated with graphon. Finally, we describe an operational algorithm that finds stage-wise graphs that outperform manually designed DenseNet as well as randomly wired architectures in [32].

Table 1: Performance (accuracy and standard deviation) on ImageNet and ImageNet V2.

| Model | # Param | ImageNet Val % | | ImageNet V2 Test % | |
|---|---|---|---|---|---|
| | | Top-1 | Top-5 | Top-1 | Top-5 |
| DenseNet-121 | 7.98 M | $75.39 \pm 0.07$ | $92.51 \pm 0.05$ | $62.92 \pm 0.06$ | $84.36 \pm 0.03$ |
| Random Deletion | 8.02 M | $75.30 \pm 0.09$ | $92.48 \pm 0.07$ | $63.41 \pm 0.31$ | $84.74 \pm 0.13$ |
| WS-G (4, 0.75) | 8.04 M | $75.43 \pm 0.11$ | $92.54 \pm 0.05$ | $63.62 \pm 0.18$ | $84.56 \pm 0.11$ |
| NAS Graphon | 7.94 M | $\mathbf{75.47} \pm 0.12$ | $\mathbf{92.57} \pm 0.04$ | $\mathbf{63.74} \pm 0.08$ | $\mathbf{84.71} \pm 0.24$ |
| DenseNet-169 | 14.15M | $76.74 \pm 0.02$ | $93.28 \pm 0.06$ | $65.29 \pm 0.29$ | $85.90 \pm 0.21$ |
| WS-G (4, 0.75) | 14.23M | $76.94 \pm 0.06$ | $93.37 \pm 0.07$ | $65.18 \pm 0.23$ | $85.79 \pm 0.13$ |
| NAS Graphon | 14.14M | $\mathbf{77.14} \pm 0.16$ | $\mathbf{93.44} \pm 0.09$ | $\mathbf{65.36} \pm 0.25$ | $\mathbf{85.96} \pm 0.14$ |
| DenseNet-201 | 20.01M | $77.42 \pm 0.06$ | $93.60 \pm 0.04$ | $65.84 \pm 0.17$ | $86.04 \pm 0.09$ |
| WS-G (4, 0.75) | 20.10M | $77.56 \pm 0.11$ | $93.67 \pm 0.07$ | $65.99 \pm 0.09$ | $86.29 \pm 0.15$ |
| NAS Graphon | 19.94M | $\mathbf{77.82} \pm 0.14$ | $\mathbf{93.80} \pm 0.09$ | $\mathbf{66.44} \pm 0.15$ | $\mathbf{86.59} \pm 0.38$ |
| DenseNet-264 | 33.34M | $77.99 \pm 0.02$ | $93.76 \pm 0.07$ | $66.11 \pm 0.11$ | $86.35 \pm 0.32$ |
| WS-G (4, 0.75) | 33.42M | $78.13 \pm 0.13$ | $93.93 \pm 0.09$ | $66.82 \pm 0.28$ | $86.54 \pm 0.24$ |
| NAS Graphon | 33.26M | $\mathbf{78.34} \pm 0.07$ | $\mathbf{94.06} \pm 0.04$ | $\mathbf{66.91} \pm 0.16$ | $\mathbf{86.76} \pm 0.14$ |

Although we find neural architectures with good performance, we remind the reader that absolute performance is not the goal of this paper. Future work involves expanding the work to different operators in the same stage graph. This can be achieved, for example, in the same manner that digraphon accommodates different types of connections. We contend that the results achieved in this paper should not be considered an upper bound, but only the beginning, of what can be achieved. We believe this work opens the door toward advanced NAS algorithms in the space of graphon and the cut distance metric.

## A    IMPLEMENTATION DETAILS

### A.1    THE GLOBAL STRUCTURE AND THE CELL STRUCTURE OF DENSENET

The DenseNet network contains a stem network before the first stage, which contains a $3 \times 3$ convolution, batch normalization, ReLU and max-pooling. This is followed by three stages for CIFAR-10 and four stages for ImageNet. Between every two stages, there is a transition block containing a $1 \times 1$ convolution for channel reduction and a $2 \times 2$ average pool with stride 2 for downsampling. The network ends with a $7 \times 7$ global average pooling and a linear layer before a softmax.

Figure 4 shows the cell structure for DenseNet, which contains two convolutions with different kernel size: $1 \times 1$ and $3 \times 3$. Each of the two convolutions are immediately preceded by a batch normalization and ReLU. Every cell in the same stage outputs $c$ channels. The input to the $n^{\text{th}}$ cell is the concatenation of outputs from the cell 1 to cell $n - 1$, for a total of $c(n - 1)$ channels. As every cell increments the number of input channels by $c$, it is called the growth rate.

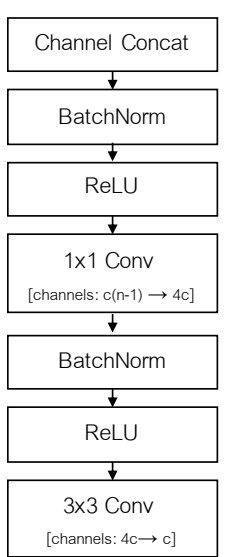

Figure 4: The DenseNet cell structure used in all experiments.

### A.2    HYPERPARAMETERS

Some detailed hyperparameters in our experiments are as follows.

During the architecture search stage, we train for 300 epochs. The learning rate schedule follows DenseNet, i.e. The initial learning rate is set to 0.1, and is divided by 10 at epochs 150 and 225. Momentum is set at 0.9. Our batch size is 64 and growth rate in the DenseNet framework is set to 32. For the Gumble softmax, initial temperature is set at 1.0 and the minimum temperature is set at 0.1 with an anneal rate of 0.03.

For ImageNet training, we train for 90 epochs. We use label smoothing, which assigns probability 0.1 to all labels other than the ground truth. We use a batch size of $256 = 64$ per GPU $\times$ 4 GPUs or $260 = 52$ per GPU $\times$ 5 GPUs, depending on the GPU memory size. For the 4 stages, growth rates are set at 26,26,26,32 to match the number of parameters for Densenet-121. Learning rate is set at 0.1, divided by 10 at epochs 30, 60, and 80. We use Nesterov momentum of 0.9 and weight decay of 0.00005.

### A.3    DATA AUGMENTATION

We also adopt the following data augmentation for ImageNet training, which are executed sequentially: Inception-style random cropping and aspect ratio distortion, resizing the image to $224 \times 224$, color jitter with factors for brightness, contrast and saturation all set to 0.4, horizontal flip with probability 0.5, and AlexNet-style random changes to the brightness of the RGB channels [15].

## B    THE SEARCH ALGORITHM

Algorithm 1 shows the detailed procedures. Specifically, for every node $v$ on the graph, we sample $K$ subsets of nodes that could provide input to $v$, while avoiding cycles (lines 5-7). For example, for the seventh node $v_7$, we could sample $\{v_1, v_3, v_5\}$ or $\{v_4, v_6\}$ and so on. We assign one weight parameter $\pi_k$ to every input subset (line 8). The model parameters for node $u$ is denoted by $\boldsymbol{\theta}_u$. Every sampled input edge subset $\mathbb{U}(v, k)$ employ the same neural network operations in $u$, but with different parameters $\boldsymbol{\theta}_{u,k}$ (lines 9-11). In particular, the input to node $v$ is the convex combination

---

**Algorithm 1** Search for Graphon

---

1: **procedure** INITIALIZE($\mathbb{V}$, $k$)
2:     Input: totally ordered nodes $\mathbb{V}$, number of subsets $k$
3:     Output: $\mathbb{I}^s(v)$, $\boldsymbol{\theta}_{u,i}$, $\boldsymbol{\alpha}_v$, $\mathbb{U}_i$
4:     **for each** node $v \in V$ **do**
5:         $\mathbb{I}(v) \leftarrow \{u \mid u \prec v, u \in \mathbb{V}\}$                 $\triangleright$ all nodes preceding $v$ to avoid cycles
6:         $\mathbb{I}^p(v) \leftarrow \mathcal{P}(\mathbb{I}(v))$                       $\triangleright$ the powerset of $\mathbb{I}(v)$
7:         $\mathbb{I}^s(v) \leftarrow$ a random $K$-element subset of $\mathbb{I}^p(v)$
8:         $\boldsymbol{\alpha}_v \leftarrow$ a random vector in $\mathbb{R}^K$            $\triangleright$ weights for the subsets
9:         **for each** input subset $\mathbb{U}(v,k) \in \mathbb{I}_s(v)$ **do**
10:             **for each** node $u \in \mathbb{U}(v,k)$ **do**
11:                 INITWEIGHT($\boldsymbol{\theta}_{u,k}$)         $\triangleright$ create a set of weights $\theta_{u,k}$ for $u$
12: **procedure** STAGEFORWARD($\mathbb{V}$, $\boldsymbol{x}$, $\tau$, $\mathbb{I}^s(v)$, $\boldsymbol{\theta}_{u,i}$, $\boldsymbol{\pi}_v$, $\mathbb{U}_i$)
13:     Input: nodes $\mathbb{V}$, input to the stage $\boldsymbol{x}$, temperature $\tau$, initialized $\mathbb{I}^s(v)$, $\boldsymbol{\theta}_{u,i}$, $\boldsymbol{\pi}_v$, $\mathbb{U}_i$
14:     Output: a feature map extracted by the current stage
15:     $in(v_{in}) \leftarrow \boldsymbol{x}$                  $\triangleright$ $v_{in}$ is a dummy sending outputs to all nodes.
16:     **for each** node $v \in V$ **do**
17:         **for each** input subset $\mathbb{U}(v,k) \in \mathbb{I}_s(v)$ **do**
18:             **for each** node $u \in \mathbb{U}(v,k)$ **do**
19:                 $out(u,k) \leftarrow f_u(in(u), \boldsymbol{\theta}_{u,k})$     $\triangleright$ apply the operation of $u$ to its input
20:             $out(k) \leftarrow$ AGGREGATE($out(u,k)$, $\forall u$)        $\triangleright$ sum or concatenation
21:         Sample $\boldsymbol{\gamma} \sim$ Gumbel(0, 1)
22:         $a_k \leftarrow \frac{\exp((\log \pi_{v,k} + \gamma_i)/\tau)}{\sum_j \exp((\log \pi_{v,j} + \gamma_j)/\tau)}$          $\triangleright$ perform Gumbel softmax
23:         $in(v) \leftarrow \sum_{k=1}^{K} a_k \, out(k)$
        **return** $in(v_{out})$         $\triangleright$ $v_{out}$ is a dummy whose input is the stage's output.

---

of the outputs from all $K$ subsets.

$$in(v) = \sum_{k=1}^{K} a_k \, out(k) \tag{6}$$

This allows all nodes in the same subset to learn to collaborate, which would be difficult if the parameters were shared across input subsets. During the forward computation, outputs from nodes in the same input subset are aggregated by either summation or concatenation (line 20). Finally, different input subsets are forced to compete by the Gumbel softmax (lines 21-23). We pick the input edge subset with the highest $\pi$ as the winning adjacency matrix.

## C   THE CUT DISTANCE $\delta_\square$

For a partition $\mathbb{S}$, $\mathbb{T}$ of $\mathbb{V}$, the cut size is defined as

$$\text{cut}(\mathbb{S}, \mathbb{T}, \mathcal{G}) = \sum_{v_i \in \mathbb{S}, v_j \in \mathbb{T}} \alpha_i \alpha_j \beta_{ij} \tag{7}$$

When two graphs $\mathcal{G}$ and $\mathcal{G}' = (\mathbb{V}', \mathbb{E}')$ have the same set of nodes (i.e. $\mathbb{V} = \mathbb{V}'$), the partition $\mathbb{S}$, $\mathbb{T}$ applies to both graphs. We can then define

$$d_\square(\mathcal{G}, \mathcal{G}') = \max_{\mathbb{S}, \mathbb{T} \subset \mathbb{V}} \frac{1}{|\mathbb{V}|^2} \left| \text{cut}(\mathbb{S}, \mathbb{T}, \mathcal{G}) - \text{cut}(\mathbb{S}, \mathbb{T}, \mathcal{G}') \right| \tag{8}$$

When the graphs have the same number of nodes, but the correspondence between nodes is unknown, the distance is defined as the minimum over all isomorphism $\tilde{\mathcal{G}} \cong \mathcal{G}$.

$$\hat{\delta}_\square(\mathcal{G}, \mathcal{G}') = \min_{\tilde{\mathcal{G}} \cong \mathcal{G}} \max_{\mathbb{S}, \mathbb{T} \subset \mathbb{V}} d_\square(\tilde{\mathcal{G}}, \mathcal{G}') \tag{9}$$

In general, the two graphs do not have the same number of nodes. Thus, in the most general metric $\delta_\square$, we allow the correspondence between nodes to be fractional. We define the "overlay" matrix $L \in \mathbb{R}^{|\mathbb{V}| \times |\mathbb{V}'|}$. The entry $L_{ip}$ denotes the fractional mapping from $u_i \in \mathbb{V}$ to $u_p \in \mathbb{V}'$, subject to

$$\sum_{p=1}^{|\mathbb{V}'|} L_{ip} = \alpha_i(\mathcal{G}) \text{ and } \sum_{i=1}^{|\mathbb{V}|} L_{ip} = \alpha_i(\mathcal{G}') \tag{10}$$

Based on $L$, we can create two new graphs, $\mathcal{G}[L]$ and $\mathcal{G}'[L^\top]$, which have the same set of nodes $\mathbb{V} \times \mathbb{V}'$ and a well defined distance $d_\square(\mathcal{G}[L], \mathcal{G}'[L^\top])$. In both graphs, the weight of node $(u_i, u_p)$ is $L_{ip}$. The edge weights are different, however; the weight of edge $((u_i, u_p), (u_j, u_q))$ in $\mathcal{G}[L]$ is $\beta_{ij}$, whereas in $\mathcal{G}'[L^\top]$ it is $\beta_{pq}$. The cut distance $\delta_\square$ takes the minimum over all possible overlay matrices.

$$\delta_\square(\mathcal{G}, \mathcal{G}') = \min_{L \in \mathcal{L}} d_\square(\mathcal{G}[L], \mathcal{G}'[L^\top]) \tag{11}$$

The fractional overlay is applicable even when the two graphs have the same node count and can lead to a lower distance than integer overlay.

## D  Fractional Upsampling

In fractional upsampling, we start with a graph $\mathcal{G}$ with $n$ nodes and generate a new graph with $n+m$ nodes $(0 < m < n)$. To achieve this, we perform two operations sequentially. First, we shift the node weights such that the $m$ nodes have weight $2/(n+m)$ and the rest $n-m$ nodes have weights $1/(n+m)$. Next, in what we call a partial blow-up, we split each of the $m$ nodes into two nodes. After the upsampling operation, all nodes have equal weights $1/(n+m)$. The requirement for equal weight is needed as the nodeweights are the probabilities of every node being sampled.

Here, the $k$-way split of node $v_i$ is defined as replacing $v_i$ with $k$ new nodes $v_{i1}, \ldots, v_{ik}$, each having node weight $\alpha_i/k$. For any other node $v_j$, the edge $(v_{ik}, v_j)$ has the same weight as $(v_i, v_j)$ and the same applies to $(v_j, v_{ik})$.

In the following, we analyze the cut distance between the new graph and the original graph and show its upper bound is $\frac{(n-m)m}{n(n+m)}\beta_\Delta$, where $\beta_\Delta$ denotes the maximum difference between any two edgeweights.

### D.1  Weight Shifting

**Theorem 3.** *Let $\mathcal{G} = (\mathbb{V}, \mathbb{E})$ and $\mathcal{G}' = (\mathbb{V}, \mathbb{E})$ be two weighted graphs that have the same set of $n$ nodes with different nodeweights and the same set of edges with the same edgeweights. If every node in $\mathcal{G}$ has nodeweight $1/n$, whereas $\mathcal{G}'$ has $m$ nodes $(0 < m < n)$ with nodeweight $2/(n+m)$ and $n-m$ nodes having weight $1/(n+m)$, then*

$$\delta_\square(\mathcal{G}, \mathcal{G}') \leq \beta_\Delta \frac{(n-m)m}{n(n+m)}. \tag{12}$$

*where $\beta_\Delta$ denotes the maximum difference between any two edge weights.*

*Proof.* We create an overlay matrix $L$ with the diagonal terms $L_{ii} = min(\alpha_i(\mathcal{G}), \alpha_i\mathcal{G}')$. For the first $m$ diagonal terms, $L_{ii} = 1/n$, and for the rest the diagonal terms are $1/(n+m)$. Then

$$\delta_\square(\mathcal{G}, \mathcal{G}') \leq d(\mathcal{G}, \mathcal{G}') \leq \sum_{i,j,p,q} L_{ip} L_{jq}(\beta_{ij}(G) - \beta_{pq}(G)). \tag{13}$$

If $i = p$ and $j = q$, then $\beta_{ij}(G) = \beta_{pq}(G)$. Thus,

$$\sum_{i,j,p,q} L_{ip} L_{jq}(\beta_{ij}(G) - \beta_{pq}(G)) = \beta_\Delta \sum_{i \neq p \text{ or } j \neq q} L_{ip} L_{jq}$$

$$= \beta_\Delta \left(1 - \left(\sum_i L_{ii}\right)^2\right) = \beta_\Delta \frac{(n-m)m}{n(n+m)} \tag{14}$$

∎

### D.2 PARTIAL BLOW-UP

**Theorem 4.** *Let $\mathcal{G} = (\mathbb{V}, \mathbb{E})$ be a weighted graph without self-loops and $\mathcal{G}'$ be the graph resulted from the $k$-way split of a node $v_i \in \mathbb{V}$, $\delta_\square(\mathcal{G}, \mathcal{G}') = 0$.*

*Proof.* We construct the overlay matrix $L$ as follows. $L_{jj} = 1/n, \forall j \neq i$. $L_{ip} = 1/kn$, for $p = 1, \ldots, k$. All other entires in $L$ are zeros.

Thus, the graphs $\mathcal{G}[L]$ and $\mathcal{G}'[L^\top]$ have the same set of nodes with non-zero weights, which include $(u_j, u_j)$ for $j \neq i$, and $(u_i, u_{ip})$ for $p = 1, \ldots, k$. The nodes $(u_j, u_j)$ have weight 1 and the nodes $(u_i, u_{ip})$ have weight $1/k$. The rest of the nodes have weight 0 and do not enter the computation of $\delta_\square$.

In both $\mathcal{G}[L]$ and $\mathcal{G}'[L^\top]$, for all $j, j' \neq i$, the edge weight between $(u_j, u_j)$ and $(u'_j, u'_j)$ is the same weight in the original graph $\beta_{jj'}(\mathcal{G})$. Additionally, in both $\mathcal{G}[L]$ and $\mathcal{G}'[L^\top]$, the edge weight between $(u_j, u_j), j \neq i$ and $(u_i, u_{ip}), p = 1, \ldots, k$ is $\beta_{ij}(\mathcal{G})$. In $\mathcal{G}[L]$, there are no self loops, so $\beta_{ii}(\mathcal{G}) = 0$. Similarly, in $\mathcal{G}'[L^\top]$, there are no edges between $u_{ip}$ and $u_{iq}$, so $\beta_{ip,iq}(\mathcal{G}') = 0$. Therefore, the graphs $\mathcal{G}[L]$ and $\mathcal{G}'[L^\top]$ are identical and $d_\square(\mathcal{G}[L], \mathcal{G}'[L^\top]) = 0$.

Since the metric $\delta_\square$ is non-negative, $\delta_\square(\mathcal{G}[L], \mathcal{G}'[L^\top]) = \min_{L \in \mathcal{L}} d_\square(\mathcal{G}[L], \mathcal{G}'[L^\top]) = 0$. ∎

**Corollary 1.** *Let $\mathcal{G} = (\mathbb{V}, \mathbb{E})$ be a weighted graph and $\mathcal{G}'$ be the graph resulted from the $k$-way split of any subset of nodes $\mathbb{V}_s \subseteq \mathbb{V}$, then $\delta_\square(\mathcal{G}, \mathcal{G}') = 0$.*

*Proof.* $\mathcal{G}'$ can be realized by splitting the nodes in $\mathbb{V}_s \subseteq \mathbb{V}$ sequentially. Every split keeps the distance with the previous graph at zero. ∎

**Corollary 2.** *Let $\mathcal{G} = (\mathbb{V}, \mathbb{E})$ and $\mathcal{G}[k]$ be a weighted graph and its $k$-way blow-up, respectively, then $\delta_\square(\mathcal{G}, \mathcal{G}[k]) = 0$.*

### D.3 COMPARING K-FOLD BLOW-UP WITH INTERPOLATION

Another possible approach for upsampling of a weighted graph is interpolating its adjacency matrix. We analyze the distance between the interpolated graph w.r.t. the blow-up graph and show interpolation results in higher distance under moderate assumptions.

In this analysis, we adopt the following notations and definitions. Let $\mathcal{G}$ denote a weighted graph with $n$ nodes. The adjacency matrix of $\mathcal{G}$ is upper-triangular and has an all-zero diagonal. This is the same as the graphon parameterization that we adopt in this paper. Further, let $\mathcal{G}[k]$ be the k-fold blow-up of $\mathcal{G}$ and let $\mathcal{G}^{itpl}[k]$ be a graph obtained by interpolating the adjacency matrix of $\mathcal{G}$ to $kn$ nodes. In all three graphs, let the sum of nodeweights be 1 and evenly distributed among all nodes.

We focus on the analysis of the one-dimensional linear interpolation, where $\beta_{ij}(\mathcal{G}^{itpl}[k])) = (i/k - \lfloor i/k \rfloor)\beta_{\lfloor i/k \rfloor \lfloor j/k \rfloor}(\mathcal{G}) + (\lceil i/k \rceil - i/k)\beta_{\lceil i/k \rceil \lfloor j/k \rfloor}(\mathcal{G})$. For example, if we interpolate a $3 \times 3$ matrix to the size $3k \times 3k$, we would first repeat every row $k$ times and interpolate horizontally. More specifically, the row $[a, b, c]$ will become $[a, \frac{(k-1)a+b}{k}, \ldots, \frac{a+(k-1)b}{k}, b, \frac{(k-1)b+c}{k}, \ldots, \frac{b+(k-1)c}{k}, c, \frac{(k-1)c}{k}], \ldots, \frac{c}{k}]$. We note the row sum is $\frac{k+1}{2}a + kb + kc$.

**Theorem 5.** *If (1) the adjacency matrix of $\mathcal{G}$ contains a column with non-zero sum and (2) $\mathcal{G}^{itpl}[k]$ and $\mathcal{G}[k]$ are optimally overlaid (i.e. $\hat{\delta}_\square(\mathcal{G}[k], \mathcal{G}^{itpl}[k]) \leq \hat{\delta}_\square(\mathcal{G}[k], \mathcal{G}') \forall \mathcal{G}' \simeq \mathcal{G}^{itpl}[k]$), then $\delta_\square(\mathcal{G}, \mathcal{G}^{itpl}[k]) > \delta_\square(\mathcal{G}, \mathcal{G}[k])$.*

*Proof.* From Theorem 2.3 of [3], we have the inequality

$$\left(\frac{\hat{\delta}_\square\left(\mathcal{G}[k], \mathcal{G}^{itpl}[k]\right)}{32}\right)^{67} \leq \delta_\square\left(\mathcal{G}[k], \mathcal{G}^{itpl}[k]\right) \tag{15}$$

Note that $\mathcal{G}[k]$ and $\mathcal{G}[\text{itpl}(k)]$ have equal number of nodes. By definition, for a partition $\mathbb{S}, \mathbb{T}$ of the node set $\mathbb{V}$,

$$
\begin{aligned}
&\hat{\delta}_\square\left(\mathcal{G}[k], \mathcal{G}^{\text{itpl}}[k]\right) \\
&= \max_{\mathbb{S},\mathbb{T}} \left|\text{cut}(\mathbb{S}, \mathbb{T}, \mathcal{G}[k]) - \text{cut}(\mathbb{S}, \mathbb{V}, \mathcal{G}^{\text{itpl}}[k])\right| \\
&= \max_{\mathbb{S},\mathbb{T}} \left| \sum_{v_i \in \mathbb{S}, v_j \in \mathbb{T}} \alpha_i(\mathcal{G}[k])\alpha_j(\mathcal{G}[k])\beta_{ij}(\mathcal{G}[k]) - \alpha_i(\mathcal{G}^{\text{itpl}}[k])\alpha_j(\mathcal{G}^{\text{itpl}}[k])\beta_{ij}(\mathcal{G}^{\text{itpl}}[k]) \right| \quad (16) \\
&= \max_{\mathbb{S},\mathbb{T}} \frac{1}{k^2 n^2} \left| \sum_{v_i \in \mathbb{S}, v_j \in \mathbb{T}} (\beta_{ij}(\mathcal{G}[k]) - \beta_{ij}(\mathcal{G}^{\text{itpl}}[k])) \right|
\end{aligned}
$$

Now we construct the partition $\mathbb{S}', \mathbb{T}'$ such that the first $km$ nodes belong to $\mathbb{S}'$ and the rest $kn - km$ nodes belong to $\mathbb{T}'$. Since the adjacency matrix of $\mathcal{G}$ is upper triangular and non-zero, without loss of generality we pick $m$ such that $\sum_{i=1}^{m} \beta_{im}(\mathcal{G}) \neq \sum_{i=m+1}^{n} \beta_{im}(\mathcal{G})$. Since all entries below the diagonal are zero, that inequality is satisfied as long as the column sum at the index $m$ is non-zero. Hence, we let $\mathbb{S}' = \{v_1 \ldots v_{km}\}$ and $\mathbb{T}' = \{v_{km+1}, \ldots, v_{kn}\}$. Let the cut distance under this partition $\mathbb{S}', \mathbb{T}'$ be $\tilde{\delta}\left(\mathcal{G}[k], \mathcal{G}^{\text{itpl}}[k]\right)$, then

$$
\begin{aligned}
\tilde{\delta}\left(\mathcal{G}[k], \mathcal{G}^{\text{itpl}}[k]\right) &= \frac{1}{k^2 n^2} \left| \sum_{v_i \in \mathbb{S}', v_j \in \mathbb{T}'} (\beta_{ij}(\mathcal{G}[k]) - \beta_{ij}(\mathcal{G}^{\text{itpl}}[k])) \right| \\
&= \frac{1}{k^2 n^2} \left| k \sum_{i=1}^{m} \sum_{j=m+1}^{n} \beta_{ij}(\mathcal{G}) + k \sum_{i=m+1}^{n} \sum_{j=1}^{m} \beta_{ij}(\mathcal{G}) \right. \\
&\quad \left. - \sum_{i=1}^{km} \sum_{j=km+1}^{kn} \beta_{ij}(\mathcal{G}^{\text{itpl}}[k]) - \sum_{i=km+1}^{kn} \sum_{j=1}^{km} \beta_{ij}(\mathcal{G}^{\text{itpl}}[k]) \right| \quad (17) \\
&= \frac{1}{k^2 n^2} \left| \frac{k-1}{2} \left( \sum_{i=1}^{m} \beta_{im}(\mathcal{G}) - \sum_{i=m+1}^{n} \beta_{im}(\mathcal{G}) + \sum_{i=m}^{n} \beta_{i1}(\mathcal{G}) \right) \right| \\
&= \frac{1}{k^2 n^2} \left| \frac{k-1}{2} \left( \sum_{i=1}^{m} \beta_{im}(\mathcal{G}) - \sum_{i=m+1}^{n} \beta_{im}(\mathcal{G}) \right) \right| > 0
\end{aligned}
$$

Thus, $\hat{\delta}_\square\left(\mathcal{G}[k], \mathcal{G}^{\text{itpl}}[k]\right) \geq \tilde{\delta}\left(\mathcal{G}[k], \mathcal{G}^{\text{itpl}}[k]\right) > 0$. By Eq 15, $\delta_\square\left(\mathcal{G}[k], \mathcal{G}^{\text{itpl}}[k]\right) > 0$, implying $\delta_\square\left(\mathcal{G}, \mathcal{G}^{\text{itpl}}[k]\right) > 0$. Since $\delta_\square\left(\mathcal{G}, \mathcal{G}[k]\right) = 0$, we have proven the desired proposition. ∎

### D.4 DISCUSSION

Theorem 5 shows the k-fold blow-up method is a better approximation of the original graph in terms of the cut distance $\delta_\square$ than the 1D linear interpolation. But the exact k-fold blow-up is only applicable when $k$ is an integer. If a graph of size $n + m (0 < m < n)$ is desired, we need to resort to the fractional blow-up method, which has been analyzed in Theorems 3 and 4. We show that when $m$ is 1 or $n - 1$, this partial blowup operation does not cause $\delta_\square$ to change more than $O(\beta_\Delta/n)$. However, when $m$ is $n/2$, $\delta_\square$ between the original graph and the new graph could be up to $\beta_\Delta/6$. This suggests that the fractional upsampling results in a graph that is similar to the original when only a small number of nodes (relative to $n$) is added.

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
