# OpenReview forum: "Searching for Stage-wise Neural Graphs In the Limit"
_ICLR.cc/2020/Conference — Reject_

### Official Review · AnonReviewer1 · 2019-10-20
**Official Blind Review #1**

**Rating:** 3

**Review:**

The authors propose a new search space based on graphons and explore some of its benefits such as certain theoretical properties. The architecture search shares similarities with DARTS. An important difference is that the network parameters are not shared.
The paper is well-written and the authors consider that the typical reader will not be familiar with graphons. I agree that their proposed model allows for more architectures but in practice it is not much stronger than WS-G. The argumentation with respect to parameters is unclear to me. On one hand, you manually influence the number of parameters, on the other you argue that you use less parameters. Obviously, you chose that your baselines have more parameters. How do results for WS-G look like if you reduce its parameters to match yours? In fact, you were searching for an architecture on CIFAR-10 but you did not report your results here. Instead you only report your transferred results to ImageNet. Is it possible that you also report results on CIFAR-10? Finally, you do not discuss that your graph contains only one kind of node. In many NAS methods the search space contains various types of operations. Do you think this is a problem? Is there a trivial way to extend your method to cover this as well?

**Experience Assessment:**

I have published in this field for several years.

**Review Assessment: Checking Correctness Of Derivations And Theory:**

I assessed the sensibility of the derivations and theory.

**Review Assessment: Checking Correctness Of Experiments:**

I carefully checked the experiments.

**Review Assessment: Thoroughness In Paper Reading:**

I read the paper thoroughly.

---

> ### Author Response · Authors · 2019-11-15
> **Responses to reviewer #1**
>
> I agree that their proposed model allows for more architectures but in practice it is not much stronger than WS-G.
>
> We have updated results and as graph sizes increase, performance gaps become more apparent and we go up to Densenet 264 where connectivity improvements results in improvements of up to 0.8%.
>
>
> The argumentation with respect to parameters is unclear to me. On one hand, you manually influence the number of parameters, on the other you argue that you use less parameters. Obviously, you chose that your baselines have more parameters.
>
> Control over the number of parameters: The single hyperparameter we can adjust for every stage is the growth rate c. A node that has k input will have kc input channels and c output channels. Here k is determined by the randomly sampled graph (different for each of the six training sessions) and out of our control. Thus, our control over the number of parameters is imprecise. We try to match all parameters. When that's not possible, we err on giving the baselines more parameters in order to create a harsh test.
>
> How do results for WS-G look like if you reduce its parameters to match yours?
> As many newly added experiments suggest, in the range we investigated, more parameters always lead to performance improvements.
>
> Specifically, we produced two variants of WS-G that have slightly parameter counts. We report the average of six training sessions.
>
> 		                                      ImageNet-2012 Val				ImageNet V2 Test
> 	              # Param  	Top 1	Stdev	Top 5	Stdev	Top 1	Stdev	Top 5	Stdev
> WS-G 169 +	14.54M	77.11	0.06	        93.44	0.05	       65.23	0.41  	85.84	0.15
> WS-G 169	14.23M	76.94	0.06	        93.37	0.07	       65.18	0.23	        85.79	0.13
>
>
> The results show that, even reducing the parameters from 14.54 M to 14.23M has a discernible effect on the performance (a reduction of 0.17% on ImageNet and 0.05% on ImageNet V2)
>
>
> In fact, you were searching for an architecture on CIFAR-10 but you did not report your results here. Instead you only report your transferred results to ImageNet. Is it possible that you also report results on CIFAR-10?
>
> We answer this in common responses 5.
>
>
> Finally, you do not discuss that your graph contains only one kind of node. In many NAS methods the search space contains various types of operations. Do you think this is a problem? Is there a trivial way to extend your method to cover this as well?
>
> The goal of this paper is to optimize only the connections between homogeneous nodes, but each node can contain multiple different operations. As the reviewer rightly guessed, extending this to allow different operations in the same graph is possible but beyond the scope of this paper. For example, the digraphon formulation provides a way to have different types of connections in the graph. Digraphon is concerned with the direction of connections, but we can easily employ different activation functions, pooling, or any other neural operators as connections.

---

### Official Review · AnonReviewer2 · 2019-10-23
**Official Blind Review #2**

**Rating:** 1

**Review:**

This paper proposes a new graphon-based search space. Unlike most other NAS works that search for exact network structures, this paper aims to search for the random graph distribution with graphon. Overall, it provides some new angles for NAS search space design, but the experimental results are very weak.

1.  It simply ignore all other NAS works and just compares with the baseline DenseNet and random deletion/walk (WS-G). Despite that, the gain (accuracy +0.17% than DenseNet baseline) is very marginal compared to other approaches:  random-wire (accuracy +2% than resent50 baseline), FBNet (accuracy +2% than MobileNetv2 baseline).
2. According to Section 5.1, the search is performed on CIFAR-10, but there is no evaluation on CIFAR-10 at all. The only results are reported for ImageNet instead, which is kind of strange.

Given these weak results, I cannot accept this paper in the current form.

**Experience Assessment:**

I have published one or two papers in this area.

**Review Assessment: Checking Correctness Of Derivations And Theory:**

I did not assess the derivations or theory.

**Review Assessment: Checking Correctness Of Experiments:**

I assessed the sensibility of the experiments.

**Review Assessment: Thoroughness In Paper Reading:**

I read the paper at least twice and used my best judgement in assessing the paper.

---

> ### Author Response · Authors · 2019-11-15
> **Responses to reviewer #2**
>
> We thank the reviewer for useful insight and comments. Here are responses to individual questions.
>
> 1.  It simply ignore all other NAS works and just compares with the baseline DenseNet and random deletion/walk (WS-G).
>
> Most works on NAS are concerned with the structure of a single cell. After a cell is found, many cells are stacked on top of each other in order to build large-capacity models. This approach is orthogonal and complementary to our work, which is concerned with the connections among such cells. Thus, a direct comparison with these works would not provide evidence that could support or contradict our main claim.
>
> Few papers aim to optimize the stage-wise graph. This is at least partially due to the lack of methods to scale a small graph learned on small datasets to match the needs of a large dataset, which this paper provides. We did compare with an existing work that considers the stage-wise graph, which is the WS model found by the randomly wired network paper. Xie et al. (2019) showed that the WS model is competitive with several NAS works including AmoebaNet, PNAS and DARTS.
>
> Despite that, the gain (accuracy +0.17% than DenseNet baseline) is very marginal compared to other approaches:  random-wire (accuracy +2% than resent50 baseline), FBNet (accuracy +2% than MobileNetv2 baseline).
>
> As discussed in the general response (1a), we have updated the paper with more experiments with improved results (up to 0.8% over DenseNet). The main goal of the experiments is to create fair comparisons and isolate the effect of the stage-wise
>
> 2. According to Section 5.1, the search is performed on CIFAR-10, but there is no evaluation on CIFAR-10 at all. The only results are reported for ImageNet instead, which is kind of strange.
>
> As of results on CIFAR-10, recent performance improvements on are mostly achieved by regularization techniques rather than neural architecture. For this reason, we are afraid that CIFAR-10 may not have enough discriminating capability to separate different baselines. Instead, we added many more experiments, including on the newly proposed ImageNet V2 test set. Some results we have on CIFAR-10 are: 93.80% for WS and 93.93% for the graph we found.

---

### Author Response · Authors · 2019-11-15
**Common responses to all reviewers**

We thank the reviewers for valuable comments and responses.

We have uploaded a revised version of the paper including the following changes.
More extensive experiments on bigger networks and an additional test set, ImageNet V2, which provides a more accurate estimate of generalization performance. The same method on bigger graphs yields bigger performance gaps up to 0.8% over DenseNet.

Improvements in writing to further clarify our main points.

On our contribution: Most existing work on architecture transfer in NAS focus on the cell structure, which is stacked consequentially to build large networks. In this paper, we study the problem of transfering and expanding the stage-wise graph from a small dataset to a large dataset. We fill a gap in NAS research because (1) few work investigated the search for stage-wise graphs and (2) there is no known algorithm for transferring small stage-wise graphs.
To validate our approach, we applied the transfer technique on two graphs. First, We expand the WS(4, 0.25) graph, defined on 32 nodes, to the graph of 64 nodes used in Denset-264. Second, we expand the 11-node graph we found on CIFAR-10 to various DenseNet settings.  We showed that, after expansion, both maintain their performance lead over DenseNet.

The purpose of our experiment is to show that this approach is feasible and beneficial under fair comparisons. We use the same setup as much as possible across all baselines. We feel this should be encouraged as this helps in isolating the contribution of the proposed technique.

As of results on CIFAR-10, recent performance improvements on are mostly achieved by regularization techniques rather than neural architecture. For this reason, we are afraid that CIFAR-10 may not have enough discriminating capability to separate different baselines. Instead, we added many more experiments, including on the newly proposed ImageNet V2 test set. Some results we have on CIFAR-10 are: 93.80% for WS and 93.93% for the graph we found.
A small technical comment is that we improved the accuracies of the DenseNet-121 group due to improved use of the PyTorch API (switching to nn.sequential improves performance)

---

### Decision · Program_Chairs · 2019-12-19

**Decision:**

Reject

**Comment:**

This paper proposes a graphon-based search space for neural architecture search. Unfortunately, the paper as currently stands and the small effect sizes in the experimental results raise questions about the merits of actually employing such a search space for the specific task of NAS. The reviewers expressed concerns that the results do not convincingly support graphon being a superior search space as claimed in the paper.